# Extraction Method of Crack Signal for Inspection of Complicated Steel Structures Using A Dual-Channel Magnetic Sensor [note 1]

**DOI:** 10.3390/s19133001

**Published:** 2019-07-08

**Authors:** Minoru Hayashi, Taisuke Saito, Yoshihiro Nakamura, Kenji Sakai, Toshihiko Kiwa, Izumi Tanikura, Keiji Tsukada

**Affiliations:** 1Graduate School of Interdisciplinary Science and Engineering in Health Systems; Okayama University, 3-1-1 Tsushimanaka, Kita-ku, Okayama 700-8530, Japan; 2Graduate School of Natural Science and Technology; Okayama University, 3-1-1 Tsushimanaka, Kita-ku, Okayama 700-8530, Japan; 3Japan Construction Method and Machinery Research Institute, 3154 Obuchi, Fuji city, Shizuoka 417-0801, Japan

**Keywords:** eddy current testing, steel crack detection, magnetic sensor, liftoff

## Abstract

Conventional eddy current testing (ECT) using a pickup coil probe is widely employed for the detection of structural cracks. However, the inspection of conventional ECT for steel structures is difficult because of the magnetic noise caused by the nonuniform permeability of steel. To combat this challenge, we have developed a small magnetic sensor probe with a dual-channel tunneling magnetoresistance sensor that is capable of reducing magnetic noise. Applying this probe to a complicated component of steel structures—such as the welds joining a U-shaped rib and deck plate together—requires the reduction of signal fluctuation caused by the distance (liftoff) variations between the sensor probe and the subject. In this study, the fundamental crack signal and the liftoff signal were investigated with the dual-channel sensor. The results showed that the liftoff signals could be reduced and differentiated from the crack signals by the differential parameters of the dual-channel sensor. In addition, we proposed an extraction technique for the crack signal using the Lissajous curve of the differential parameters. The extraction technique could be applied to the inspections not only for flat plates but also for welded angles to detect cracks without the influence of the liftoff signal.

## 1. Introduction

Eddy current testing (ECT) using a pickup coil probe is a widely used technique employed to detect cracks in various conductive structures [1,2,3]. However, for the inspection of steel structures, crack signals are hard to discriminate from the magnetic noise caused by the nonuniform permeability of steel. Therefore, a more reliable method, which is robust against magnetic noise, is required to detect true crack signals.

Recently, nondestructive testing methods using magnetic sensors instead of a pickup coil have been reported [4,5,6]. There are advantages to the use of a magnetic sensor. Magnetic sensors can detect a wide range of frequencies with constant sensitivity. Thus, several nondestructive techniques operated at a low frequency to detect subsurface and surface cracks in various structures have been reported [7,8,9]. In addition, magnetic sensors are typically small and can detect magnetic flux density with high resolution. Therefore, the differential of local magnetic flux density can be measured more flexibly using a magnetic sensor array. Using these advantages of magnetic sensors, we have demonstrated that magnetic noise can be reduced by the differential parameters obtained by a gradiometer of a magnetic sensor [10].

However, few methods are suitable for inspection of steel complicated structures because the size and shape of typical magnetic probes do not fit the complicated components of structures. For example, the welds joining U-shaped ribs and deck plates together are difficult to inspect where fatigue cracks often appear by the vibration of vehicles. To inspect these structures, a small sensor probe of a gradiometer using dual channel magnetic sensors and an analysis method based on differential parameters are required. For this reason, we have developed a small ECT sensor probe using two tunneling magnetoresistance (TMR) sensors. In addition, the fundamental characteristics of magnetic response and data analysis for crack detection in flat steel plates have been reported [11]. However, the signal from the sensors easily fluctuates because of the distance (liftoff) variation between the sensor probe and the structure. The manual probing to maintain the liftoff is difficult because the surface of structures is often uneven and because of the liftoff change due to the fluctuation of the sensor probe when it is probed. This makes it difficult to discriminate the crack signal from the liftoff signal. Although in previous works new data processing methods and approaches have been proposed to compensate for the influence of liftoff [12,13,14], there are few methods aiming at the evaluation of complicated steel structures.

In order to show the measured signal, conventional ECT typically uses a Lissajous curve that displays the real and imaginary parts, which are produced from the change of the crack-induced signal from a single sensor output [15]. The Lissajous curve using the parameters of a single sensor is easily affected by the magnetic noise and the liftoff signal. Hence, the differential parameters of a dual-channel magnetic sensor have been used for the Lissajous curve to reduce the magnetic noise and discriminate the crack signal from the liftoff signal [16]. However, the signal fluctuation which occurs in a practical inspection of complicated structures has been poorly understood.

In this paper, the distributions of eddy current on the flat plate and magnetic field inside the induction coil were simulated to confirm that the crack signal can be detected with liftoff which will occur in the probing of the complicated structure. Then, the fundamental characteristics of the crack signal and the influence of liftoff variations were analyzed with a vertical and tilted sensor probe. As a result, an extraction technique of the crack signal using the Lissajous curve of the differential parameters obtained from a dual-channel magnetic sensor probe was proposed. In addition, the frequency dependence of the magnetic signal was also investigated to discuss the optimum frequency with which to extract the crack signal. Using the developed extraction method of the Lissajous curve, the complicated structure was measured.

## 2. Test Sample and Measurement System

A flat steel plate with artificial cracks and a U-shaped rib with artificial cracks in its welded part were prepared to evaluate the liftoff dependence, respectively. The test sample of the flat plate was a 7 mm thick SS400 with slit-like artificial cracks of different depths ranging from 0.5 to 7.0 mm (through hole). Figure 1a,b show a dual-channel sensor probe and measurement system, respectively. The sensor probe was composed of an induction coil and two TMR sensors. The two TMR sensors were installed on the long side of the rectangular induction coil with 60 turns and a cross-sectional area of 2.5 × 6.0 mm^2^. The system consisted of a voltage source, a function generator, a sensor probe with two TMRs, lock-in amplifiers, a personal computer, and an XYZ stage. The TMR sensors were driven by a voltage source applying 5 V, and the induction coil was operated by a function generator applying a sine wave voltage of 2 V_p-p_ at frequencies between 200 Hz and 5 kHz. The sensor probe was line-scanned vertical to the sample surface for the 7 mm deep slit at 0.1 mm intervals, with different liftoff between 0.1 and 0.9 mm to evaluate the liftoff dependence. It is expected that the real structures are covered by a thin paint which is uneven and cracked. In addition, for the complicated part of the structure, manual probing is required. For these reasons, we assumed the liftoff change to be between 0.1 and 0.9 mm under this condition. The sample was also line-scanned with the sensor probe tilted by 5 degrees from the vertical line. The lock-in amplifiers obtained the TMR sensor output signals as magnetic vectors with real and imaginary components, and the signals were analyzed using a computer.

Figure 2 shows the positions of cracks in the welded part of the U-shaped rib sample and the scanning path. It is well-known that cracking tends to occur along the welds. Therefore, three artificial cracks of different lengths (10, 20, and 30 mm) were prepared along the welding angle by electric discharge machining. The cracks penetrated the rib, which was 6 mm thick, and the welds. Each crack width was 1 mm. A sine wave voltage of 2 V_p-p_ at frequencies between 1 kHz and 5 kHz was applied to the sensor probe, which was then manually moved along the scanning path for each crack and non-crack area.

## 3. Simulation of Eddy Current and Magnetic Field Distribution

To evaluate the fundamental magnetic response and to estimate the intensity of the crack signal with different liftoff, the distributions of eddy current on the surface of a flat steel plate (SS400) and the magnetic field in an induction coil were simulated using the commercial software package JMAG (JSOL Corporation). The induction coil with the air area inside the coil was positioned at the center of the flat steel plate with a trough-hole slit of 7 mm depth, as shown in Figure 3. The resistance of induction coil was 1 Ohm. The typical permeability and conductivity of SS400 provided by JSOL were used for the simulation. The simulation was performed in the initial magnetization curve which was the linear region because the applied magnetic field was very weak. Each element was automatically divided into 0.4 mm meshes for the finite-element analysis. A sine wave voltage of 2 V_p-p_ at a frequency of 1 kHz was applied to the induction coil with different liftoff between 0.0 and 1.0 mm.

The eddy current was assumed to be induced along and under the induction coil. However, the eddy current flow was interrupted by the slit. Thus, the simulated eddy current was divided into two parts and the two parts of the eddy current flowed along the opposite of the slit edge (Figure 4). The eddy current density with 0.0 mm liftoff was large compared to that with 1.0 mm liftoff. This indicates that the intensity of magnetic flux induced by the eddy current is easily influenced by liftoff variations.

Figure 5 shows the wire frame of the simulation model (Figure 5a,b) and the simulated magnetic flux vectors inside the induction coil with different liftoff of 0.0 and 1.0 mm (Figure 5c,d). The eddy current did not flow in the slit, as shown in Figure 4. Therefore, the induced magnetic flux on the slit became locally weak. For this reason, the intensity and direction of the vector changed on the slit. When the magnetic sensor which is installed inside the induction coil is on the slit, the local signal change can be measured. The intensity of the magnetic vector with 0.0 mm liftoff was larger than that with 1.0 mm liftoff. However, the signal change could be seen at the bottom center of the induction coil, as shown in Figure 5d. Thus, the crack signal was able to be obtained even if the liftoff increased to 1.0 mm. To measure the small magnetic flux change caused by the crack, a TMR sensor is suitable for such a small intensity range of magnetic flux.

## 4. Results and Discussion

### 4.1. Crack Signal of Single Sensor

The magnetic vector $S_{1,n}$, $S_{2,n}$ of output signals (position = *n* (mm)) was able to be obtained by lock-in amplifiers using the outputs from two independent TMR sensors. The slit was located at the position of 10 mm. Figure 6 depicts the signal change for the real and imaginary part of the magnetic vectors of the 7 mm deep slit. The data was obtained by line-scanning vertically with 0.1 and 0.9 mm liftoff at a frequency of 1 kHz. The magnetization signal of ferromagnetism was large compared to the eddy current signal in the measured frequency range. Therefore, the real part was large compared to the imaginary part. The peak caused by the crack was observed in each figure. However, the baseline of both sides of the slit were not the same for each measured data and this may be considered to be caused by magnetic noise, including the nonuniform permeability. In addition, the baseline of both sensors shifted due to the liftoff variation. These results suggest that the crack signal is buried in the magnetic noise and signal fluctuation caused by the liftoff. Meanwhile, the change of baseline caused by the magnetic noise and liftoff obtained by each sensor were almost the same. Therefore, in order to reduce the magnetic noise and the influence of liftoff variations, the differential value of the dual-channel sensor was considered to be effective to extract the crack signal.

### 4.2. Liftoff Dependence of the Differential Parameters

The differential vector $S_{2,n}-S_{1,n}$ was able to be obtained from the dual-channel sensor probe. Figure 7 depicts the liftoff dependence of the differential parameters for the real part (Re(S2,n−S1,n)) and the imaginary part (Im(S2,n−S1,n)). The measurement sample was the flat plate and the frequency was 1 kHz. The sensor probe was vertically line-scanned with different liftoff between 0.1 and 0.9 mm. The peak-to-peak values of the differential parameters are also described (Figure 7c,d). The baselines of the differential parameters at both sides of the slit became almost the same compared to the single parameters, which are described in Figure 6. The magnetic noise seemed to be reduced by the differential value. The peak-to-peak values decreased as the liftoff increased. On the other hand, even when the liftoff increased, the baselines were almost zero. These results mean that the influence of the liftoff variations could be also reduced by the differential value obtained by the dual-channel sensor.

Although the influence of the liftoff variation may be reduced by the differential parameters, there is a case for not ignoring the influence of liftoff. When the sensor probe is not vertical to the surface, the baseline shifts of the two sensors are considered to be not the same because the liftoff of each sensor is different and the differential parameters are affected by the liftoff variations. Figure 8 shows the liftoff dependence of the differential vector when the sensor probe was tilted by 5 degrees. The peak-to-peak values of differential parameters are also described in Figure 8c,d. The baselines of each differential parameter shifted to zero and the peak-to-peak values decreased as the liftoff increased. This result suggests that the differential parameters can be influenced by the liftoff variations. However, the baseline shift of the real part was large compared with that of the imaginary part. This means that the Lissajous curve of the differential vector appears mainly in the X-direction when the signal fluctuates by the liftoff variations. Additionally, the crack signal is assumed to appear in a different direction.

In order to calculate the direction of the liftoff signal, Δx and Δy were defined as shown in Figure 8a,b. The gradient of the liftoff signal $\theta_{L}$ can be obtained by the following equation:(1)$$\theta_{L}=\arctan\left(\Delta y/\Delta x\right)$$

As well as $\theta_{L}$, the gradient of the crack signal $\theta_{C}$ can be obtained by dividing the peak-to-peak value of the imaginary part by that of the real part. Figure 9 shows the gradient of the crack signal $\theta_{C}$ and liftoff signal $\theta_{L}$. The same data from Figure 8 was used to calculate the gradients. In any liftoff the $\theta_{C}$ was large enough compared to the $\theta_{L}$. This result suggests that the crack signal appears in a different direction from the liftoff signal in the Lissajous curve of the differential parameters. In the next subsection, the Lissajous curve of differential parameters is described.

### 4.3. Lissajous Curve of the Differential Vector

In this subsection we investigated a signal extraction method to emphasize the crack-induced signal change that is not influenced by the liftoff variations caused by the probe tilt. We observed that the differential parameters of the two sensors can reduce the magnetic noise and influence of the liftoff. In addition, even when the differential parameters are affected by the liftoff, the Lissajous curves of the crack and non-crack signal are assumed to show differences. Figure 10 shows the Lissajous curves of the differential vector with different liftoff between 0.1 and 0.9 mm. The sensor probe was line-scanned vertically and not vertically at a frequency of 1 kHz. The Lissajous curves of non-crack area which correspond to the baselines shown in Figure 7a,b and Figure 8a,b are also described. When the sensor probe was vertical to the flat steel plate, the signals of the non-crack area were almost at the origin (Figure 10a). This result indicates that the influence of the liftoff variation could be reduced by the differential parameters. Thus, the crack signal was easily extracted by the Lissajous curve (Figure 10b). On the other hand, when the sensor probe was not vertical, the signal of the non-crack area shifted mainly in the X-direction as shown by the dotted arrow (Figure 10c). However, the crack signal was observed in a different direction (Figure 10d). This result suggests that the crack signal can be discriminated using the Lissajous curve even when the signal fluctuates due to the liftoff variations of the tilted probe. However, the difference between the crack signal and the signal change caused by the liftoff is assumed to depend on the frequency of the applied magnetic field. Thus, in the next subsection, the frequency dependence of each signal is discussed.

### 4.4. Frequency Dependence of the Crack Signal

The peak-to-peak values of the differential parameters at frequencies between 200 Hz and 5 kHz with the vertical and tilted sensor probe are shown in Figure 11. In regards to the peak-to-peak values of each liftoff, the real part decreased while the imaginary part increased as the frequency increased. Considering that the real part was easily affected by the liftoff compared to the imaginary part, it was expected that high frequency would be suitable to discriminate the crack signal. The peak-to-peak values of the vertical line-scan were large compared to those of the tilted line-scan. However, the gradients of the crack signal $\theta_{C}$ were assumed to be almost the same.

Figure 12 shows the gradients of the crack signal $\theta_{C}$. There is almost no difference in $\theta_{C}$ between the vertical line-scan and the titled line-scan. This means that the crack signal appears in a certain direction regardless of the tilt of the sensor probe. In regards to the gradient of each liftoff, the gradient increased as the frequency increased. This result indicates that the difference between the crack signal and the signal change caused by the liftoff variation of tilted probing becomes apparent and the crack signal can be easily distinguished from the signal change due to the liftoff. Thus, the applied field of high frequency is considered to be needed for the extraction method.

However, also of consideration is that the gradient of the liftoff signal also increases as the frequency increases, which makes it difficult to extract the crack signal. To confirm these effects, the subtracted gradient $\theta_{C}-\theta_{L}$ which was the angle formed by the crack and liftoff signal was calculated, as seen in Figure 13. The data was obtained by tilted line-scan. Even if the gradient of the liftoff increased with frequency, the gradient of the crack signal was large enough to discriminate. Consequently, the subtracted gradient at 5 kHz was maximum and this frequency was optimum in the range of frequency between 200 Hz and 5 kHz. This result suggests that the crack signal can be clearly extracted using the Lissajous curve even when the liftoff variation occurs due to the manual probing of complicated structures. In the next subsection, the measurement of the crack signal in the complicated structures is demonstrated using the developed method.

### 4.5. Steel Cracks in A Complicated Structure

Figure 14 shows Lissajous curves of the differential vector. The data was obtained by manual probing of a U-shaped rib sample with different-length cracks and measured at frequencies between 1 kHz and 5 kHz. The signal change of the non-crack area was observed (Figure 14a); this change was caused by liftoff variation which occurred during the manual probing. The lines of liftoff signal were decided as shown in Figure 14a. The gradients of the line were the $\theta_{L}$, which increased as the frequency increased. Meanwhile, the Lissajous curves of each crack are shown in the Figure 14b–d. The crack signal appeared in a different direction from that of the liftoff signal and the signal intensity of this direction may be considered to be almost the same for the different lengths of slit. This is because all the cracks were longer than the thickness of a side of the induction coil, the eddy current flowed along the induction coil, and the local signal change was measured. On the other hand, the signal intensity which appeared in the direction of the liftoff signal line was different. This difference indicates that the liftoff change occurred due to manual probing and the liftoff variation for the 30 mm crack can be considered to be large. Hence, it was possible to discriminate the crack signal from the non-crack signal using the Lissajous curve. Therefore, if the signals of non-crack area including the liftoff effect are first measured, the signal change caused by the cracks can be detected and the proposal method is useful for the actual inspection of complicated steel structures.

## 5. Conclusions

In this work, an extraction method for the detection of cracks in complicated structures was developed. It was shown by simulation and measurement that crack signals could be obtained with a small sensor probe using a dual-channel TMR sensor and that the differential parameters reduced the influence of the liftoff signal. The Lissajous curves of the differential parameters clearly differentiated between the crack signal and liftoff signal, which occur when probing the complicated structure. Hence, the extraction technique can be applied to inspections not only for welds but also for other complicated steel structures to detect cracks reliably.

## Figures and Tables

**Figure 1 sensors-19-03001-f001:**
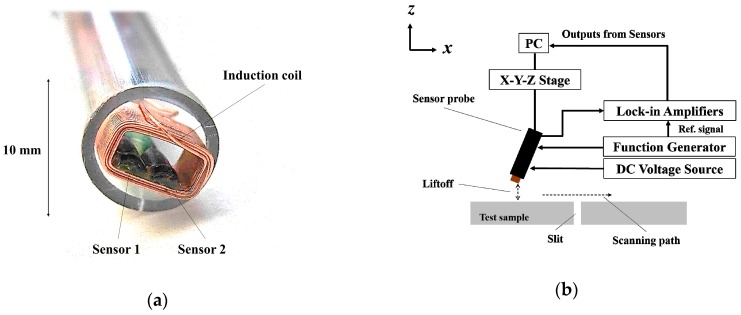
(**a**) The dual-channel sensor probe; (**b**) diagram of the measurement system.

**Figure 2 sensors-19-03001-f002:**
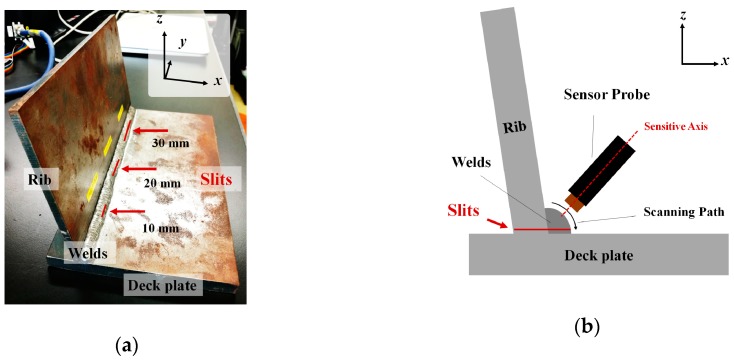
(**a**) Photo of U-shaped rib sample with different-length slits; (**b**) schematic of scanning path.

**Figure 3 sensors-19-03001-f003:**
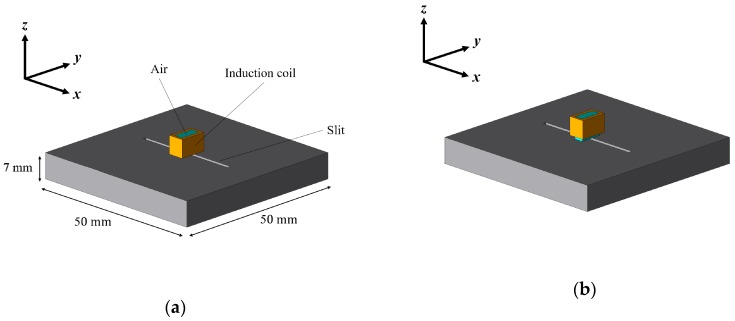
Simulation model for estimating the fundamental magnetic response with different liftoff: (**a**) with 0.0 mm liftoff; (**b**) with 1.0 mm liftoff.

**Figure 4 sensors-19-03001-f004:**
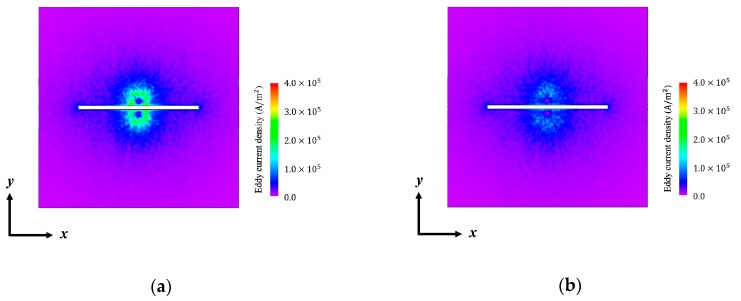
Simulation result of the eddy current density on the surface of the flat plate with different liftoff: (**a**) with 0.0 mm liftoff; (**b**) with 1.0 mm liftoff.

**Figure 5 sensors-19-03001-f005:**
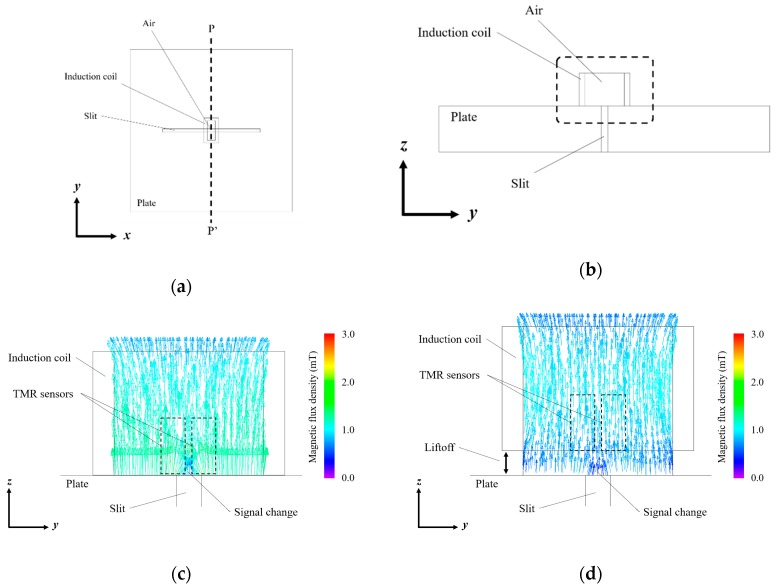
(**a**) Wire frame of the simulation model and cutting line of P to P’; (**b**) cross-sectional view of the cutting line; (**c**) simulation result of magnetic vector inside the induction coil with 0.0 mm liftoff; (**d**) simulation result of magnetic vector inside the induction coil with 1.0 mm liftoff. Legend: TMR, tunneling magnetoresistance.

**Figure 6 sensors-19-03001-f006:**
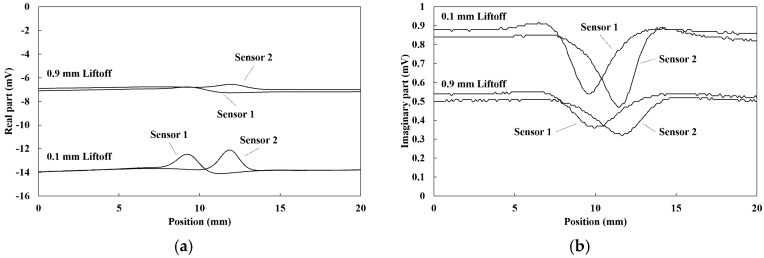
Signal changes of 7 mm deep slit between 0.1 and 0.9 mm liftoff at a frequency of 1 kHz: (**a**) real part; (**b**) imaginary part.

**Figure 7 sensors-19-03001-f007:**
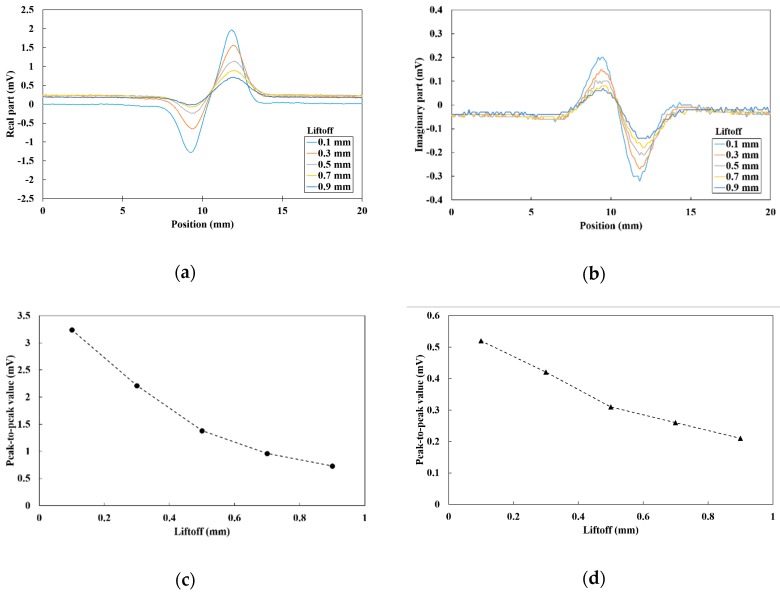
Vertical line-scan results of the 7 mm deep crack in the flat plate at a frequency of 1 kHz: (**a**) real part of the differential vector; (**b**) imaginary part of the differential vector; (**c**) liftoff dependence of the peak-to-peak real part values; (**d**) liftoff dependence of the peak-to-peak imaginary part values.

**Figure 8 sensors-19-03001-f008:**
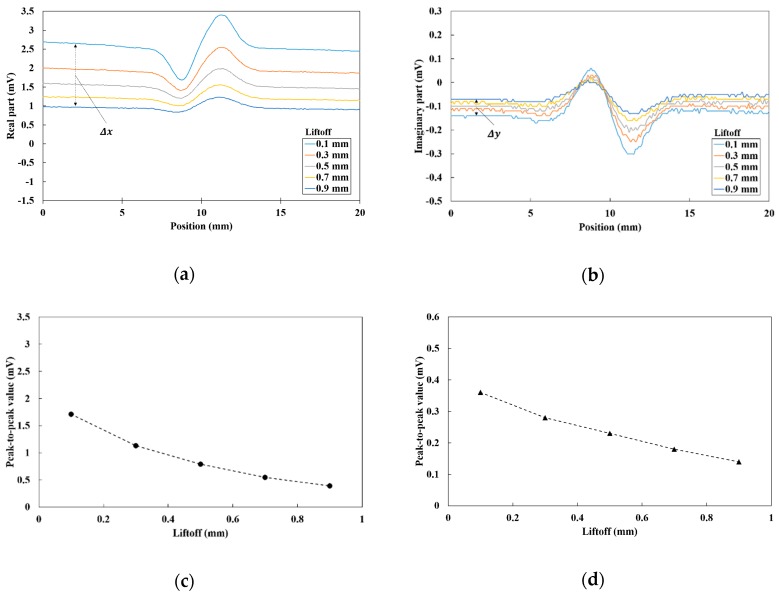
Tilted line-scan results of the 7 mm deep crack in the flat plate at a frequency of 1 kHz: (**a**) real part of the differential vector; (**b**) imaginary part of the differential vector; (**c**) liftoff dependence of the peak-to-peak real part values; (**d**) liftoff dependence of the peak-to-peak imaginary part value.

**Figure 9 sensors-19-03001-f009:**
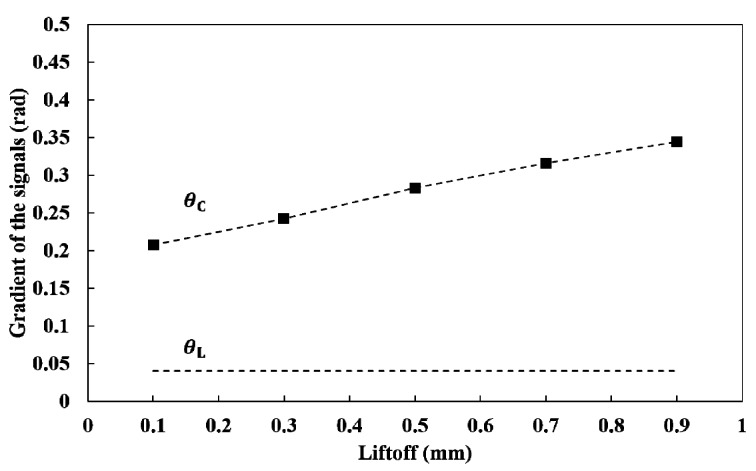
Comparison of the gradients between the crack signal $\theta_{C}$ and liftoff signal $\theta_{L}$.

**Figure 10 sensors-19-03001-f010:**
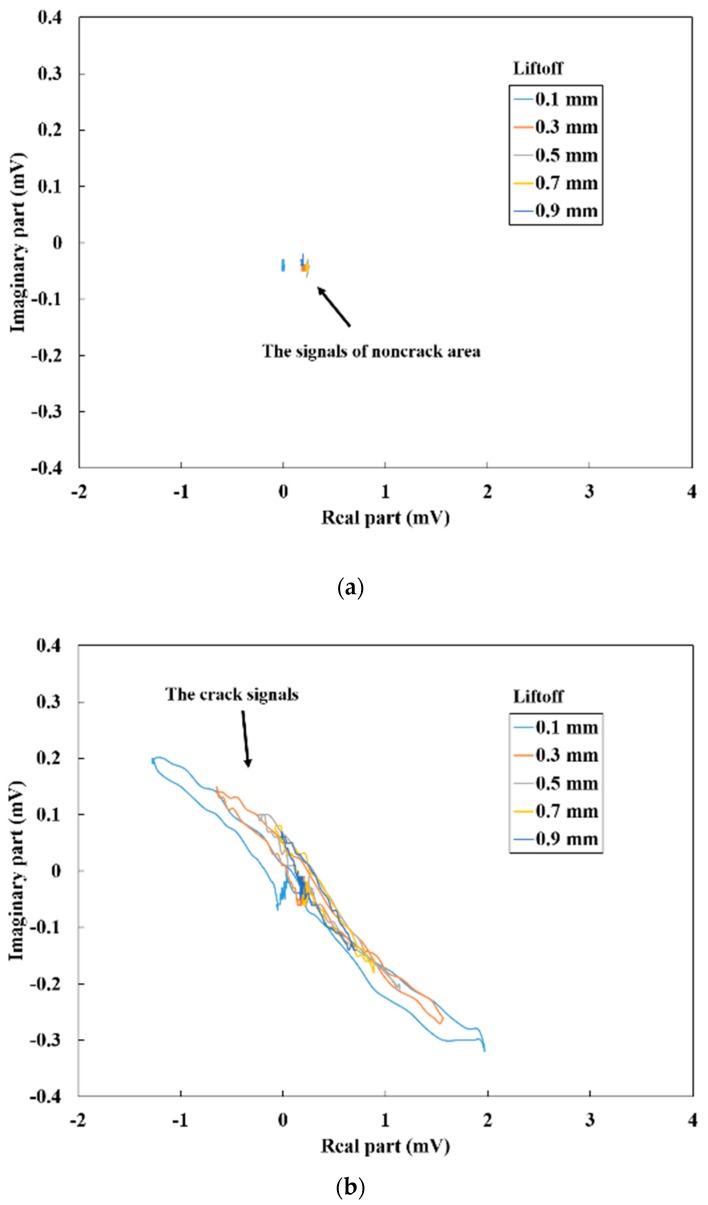
Lissajous curves of the differential vector at a frequency of 1 kHz: (**a**) vertical line-scan result of the non-crack area; (**b**) vertical line-scan result of the crack area; (**c**) tilted line-scan result of the non-crack area; (**d**) tilted line-scan result of the crack area.

**Figure 11 sensors-19-03001-f011:**
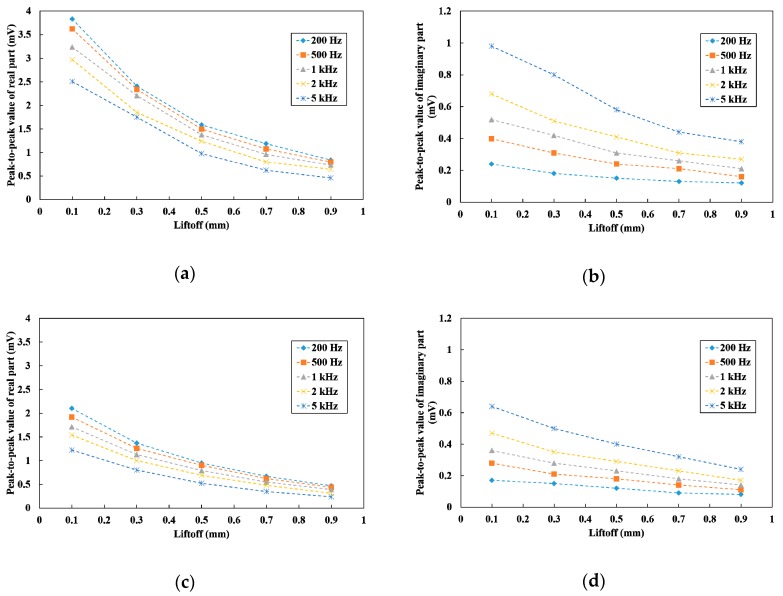
Peak-to-peak values at frequencies between 200 Hz and 5 kHz with different liftoff: (**a**) real part of the vertical line-scan; (**b**) imaginary part of the vertical line-scan; (**c**) real part of the tilted line-scan; (**d**) imaginary part of the tilted line-scan.

**Figure 12 sensors-19-03001-f012:**
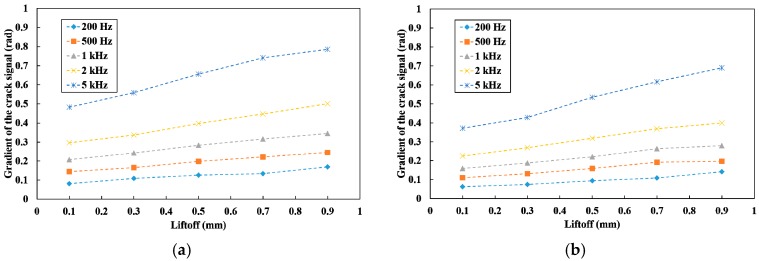
The gradient of the crack signal at frequencies between 200 Hz and 5 kHz with different liftoff: (**a**) the vertical line-scan; (**b**) the tilted line-scan.

**Figure 13 sensors-19-03001-f013:**
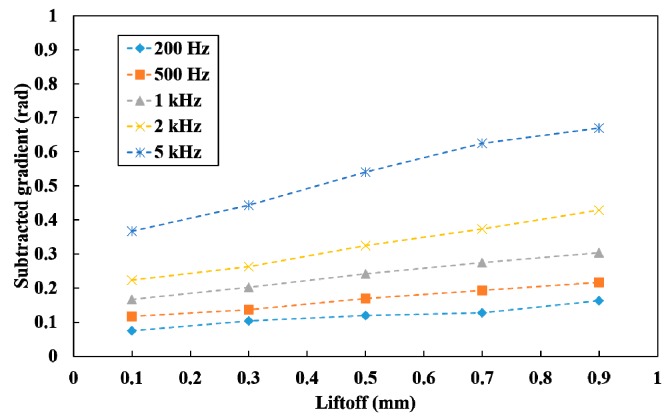
The subtracted gradient with different frequencies.

**Figure 14 sensors-19-03001-f014:**
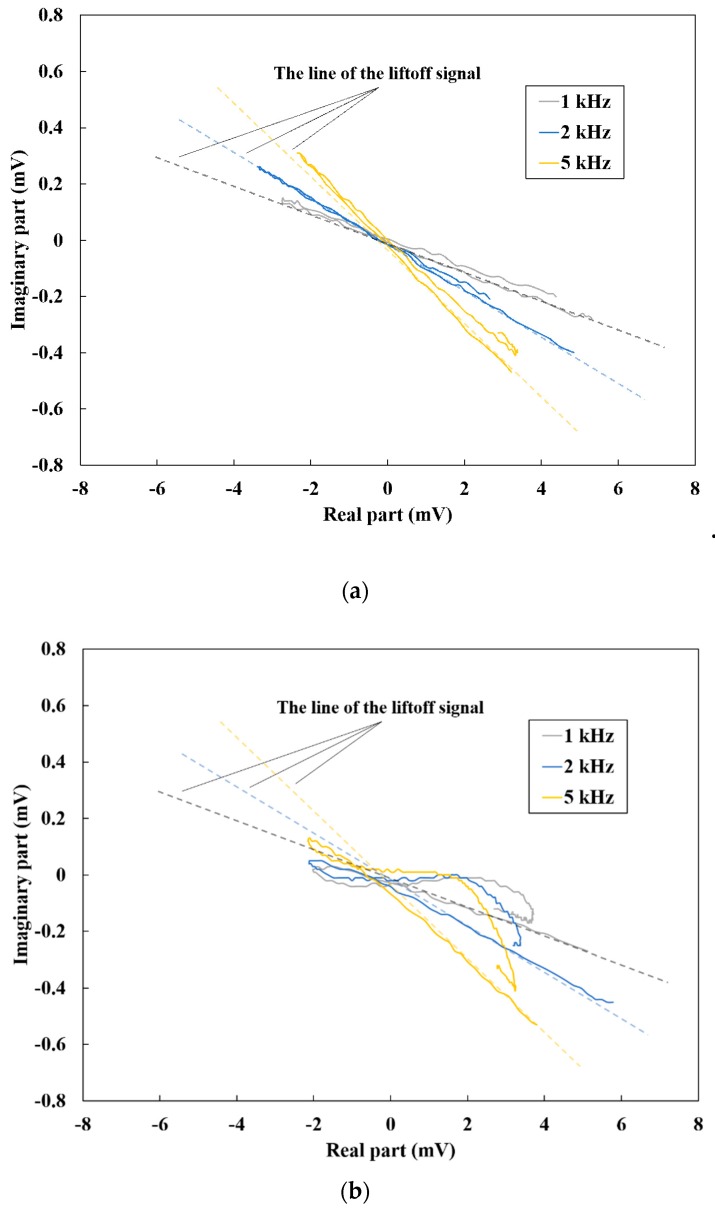
Lissajous curves of the welded part of the U-shaped rib with different-length cracks at frequencies between 1 kHz and 5 kHz: (**a**) non-crack area; (**b**) 10 mm length crack; (**c**) 20 mm length crack; (**d**) 30 mm length crack.

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
