# Peer review of "Extraction Method of Crack Signal for Inspection of Complicated Steel Structures Using A Dual-Channel Magnetic Sensor†"

_sensors, 2019, doi:10.3390/s19133001_

Reviewer 1 Report

In the INTRODUCTION line 70, the authors mentioned to estimate the optimum frequency of magnetic signal. However in the section 4.4, there is no clear statement about the optimum frequency. Pelease improve it or explain it.

Reviewer 2 Report

This paper describes that a small magnetic sensor probe with a dual-channel tunneling magnetoresistance sensor that reduce for lift off noise to detect flaws on the welds joint.

The study is interesting, and the obtained results are reasonable. My comments are as follows,

1. Please provide the figure to explain the detection principle of a flaw. It means why the intensity and gradient of the vector changed at on the slit although those of the other part of the bottom was almost the same.

2. Please provide electromagnetic parameters used in the simulation? In addition, the reviewer does not understand it is linear or nonlinear simulation.

3.  Compared with signal of real part, the signal of imaginary part was smaller by one order. Please discuss about it.

4. Please provide flaw depth and width information of flaws in the welded part of U-shaped rib sample. How did you make the flaws on the test piece.

Please discuss about the amplitude of Lissajous curve between flaw lengths of 10 mm and 20mm.

Please discuss about the shape of Lissajous curve between flaw length 30 mm and others

5. Please describe the effect the flaw tangentially located with the direction of long diameter of the coil although I understand it is out of the scope of the paper.

6. Please improve the description of conclusion.

Reviewer 3 Report

The paper presents interesting results on the development and application of an electromagnetic sensor for eddy current inspection of welds. The sections in general are well structured and text is well written. However, I would like to make a few comments bellow.

·       In the Introduction section (line 39), the text references implies that the present work is about a sensor operating in low frequencies, which is confirmed when 5khz as the maximum frequency presented. However, in the section Results and Discussion, the use of high frequency in obtaining the desired results is quoted three times (lines 209, 221 and 229). For the reader it may be difficult to understand the statement "high frequency" when substantially higher frequencies are frequently used for eddy current inspection.

·       In the Test Sample and Measurement System, section it is unclear why a frequency range between 200 Hz and 5 kHz (line 84) was used in the flat plate test sample, while a range between 1 kHz and 5 kHz (line 92) was used in the U-shaped ribbed test sample.

·       Are there any application requirements for the choice of lift off between 0.1 and 0.9 mm (line 85) in the lift off study? If there is such a motivation, I believe it would be interesting to contextualize it for the reader.

·       In the Simulation of Eddy Current and Magnetic Field Distribution, section the reader is not explained why the simulation is performed only by applying the 1 kHz frequency (line 102).

·       I wonder if the last paragraph (lines 113-122) and Figure 5 would not be a part of Results and Discussions section.

·       In the subsection Crack signal of single sensor, again it is not clear to the reader the reason for choosing the 1 kHz (line 132) as the working frequency.

·       The connection sentences (lines 198-200) between the subsections Lissajous curve of the differential vector and Frequency dependence of the crack signal have contributed a lot to the contextualization and fluidity of the text. The same was done (lines 230-233) in the subsection Steel cracks in a complicated structure. It would be an interesting exercise, and can enrich the text, make the same type of textual connection at the end of the Liftoff dependence of the differential parameter subsection.

·       The contextualization of the Lissajous curve (line 183) could be earlier in the text.

·       Figures 10 and 14 could be larger. They are difficult to see, especially when viewed in grayscale.

·       The Conclusions section is clear and assertive. However, the results generated from Simulation of Eddy Current and Magnetic Field Distribution section were not mentioned. In this way, I question again the importance for this work.

Reviewer 4 Report

Overall the paper is clearly written and well structured. It contributes to the area of nondestructive testing using eddy current method. It is well-known that the lift-off effects are the main factors affecting the performance of eddy current testing system. Literature review should be extended to include a number of previous works related to lift-off compensation techniques.

Some typos to be corrected:

l. 183  "the lilit of probe" ? -> "the tilt of probe"

l. 201  "1kHz" -> "1 kHz"

Author Response

This manuscript is a resubmission of an earlier submission. The following is a list of the peer review reports and author responses from that submission.

Round  1

Reviewer 1 Report

This article has already been published (see https://www.ndt.net/article/apwshm2018/papers/216.pdf).